# Mountain Glacier Flow Velocity Retrieval from Ascending and Descending Sentinel-1 Data Using the Offset Tracking and MSBAS Technique: A Case Study of the Siachen Glacier in Karakoram from 2017 to 2021

Qian Liang [1,2] and Ninglian Wang [1,2,3,*]

1 Shaanxi Key Laboratory of Earth Surface System and Environmental Carrying Capacity, Northwest University, Xi'an 710127, China

2 Institute of Earth Surface System and Hazards, College of Urban and Environmental Sciences, Northwest University, Xi'an 710127, China

3 State Key Laboratory of Tibetan Plateau Earth System, Environment and Resources (TPESER), Institute of Tibetan Plateau Research, Chinese Academy of Sciences, Beijing 100101, China

* Correspondence: nlwang@nwu.edu.cn

**Abstract:** Synthetic Aperture Radar images have recently been utilized in glacier surface flow velocity research due to their continuously improving imaging technology, which increases the resolution and scope of research. In this study, we employed the offset tracking and multidimensional small baseline subset (MSBAS) technique to extract the surface flow velocity of the Siachen Glacier from 253 Sentinel-1 images. From 2017 to 2021, the Siachen Glacier had an average flow velocity of 38.25 m a$^{-1}$, with the highest flow velocity of 353.35 m a$^{-1}$ located in the upper part of a tributary due to the steep slope and narrow valley. The inter-annual flow velocity fluctuations show visible seasonal patterns, with the highest flow velocity observed between May and July and the lowest between December and January. Mass balance calculated by the geodetic method based on AST14DEM indicates that the Siachen Glacier experienced a positive mass change ($0.07 \pm 0.23$ m w.e. a$^{-1}$) between 2008 and 2021. However, there was significant spatial heterogeneity revealed in the distribution, with surface elevation changes showing a decrease in the glacier tongue while thickness increased in two other western tributaries of the Siachen Glacier. The non-surface parallel flow component is correlated with the strain rate and mass balance process, and correlation analysis indicates a positive agreement between these two variables. Therefore, using glacier flow velocities obtained from the SAR approach, we can evaluate the health of the glacier and obtain crucial factors for the glacier's dynamic model. Two western tributaries of the Siachen Glacier experienced mass gain in the past two decades, necessitating close monitoring of flow velocity changes in the future to detect potential glacier surges.

**Keywords:** Siachen Glacier; glacier velocity; Synthetic Aperture Radar; offset tracking; MSBAS

## 1. Introduction

Mountain glaciers are sensitive indicators of climate change with significant global and regional climate impacts as they are part of the Cryosphere [1]. Glacier movement monitoring and research provide critical data not only for climate change research and glacier disaster prevention and control but also to understand the fundamental characteristics and movement patterns of glaciers [2]. Since the 20th century, High-Mountain Asia's glaciers have been melting rapidly as a result of global warming [3]. Despite this, there are significant regional differences in how glaciers respond to climate change; the Karakoram Mountains have glaciers that are a substantial source of freshwater for downstream communities [4]. A recent study indicates that these glaciers have sustained a slight mass gain in recent decades, which defies the evolution of the vast majority of glaciers worldwide

and has drawn attention from the scientific community [5,6]. However, due to their remote location and harsh natural environment, field observation is time-consuming and labor-intensive, making real-time and large-scale monitoring challenging. The properties of fast data acquisition, wide imaging range, rich information, and short revisit cycle associated with satellite remote sensing technology have enabled rapid and extensive monitoring of glacier changes [7–12].

The monitoring of glaciers has increasingly utilized several complex remote sensing technologies, including Synthetic Aperture Radar (SAR) [11,13], Global Navigation Satellite System (GNSS) [14–16], optical images [17–19], and unmanned aerial vehicles [20,21]. Field observations using GNSS and UAV measurement technology can obtain glacier velocity with high accuracy. However, terrain and harsh environments present challenges in achieving large-scale observations. Even for large, single glaciers, it can be difficult to obtain velocity measurements throughout the entire glacier, particularly in the upper accumulation area. Presently, research on motion in mountain glaciers primarily focuses on the tongue area, which provides accurate observation results that can verify remote sensing inversion results. Remote sensing inversion is currently the principal method for obtaining glacier velocity on a large scale. The main data sources for remote sensing inversion are optical images and SAR images. Optical images are rich in historical data, enabling long-term retrieval of glacier velocity, but are susceptible to cloud cover. SAR is an active microwave remote sensing technology that can operate under all-weather conditions and has global coverage with high spatiotemporal resolution [22]. Sentinel-1 data are the most popular SAR images presently used, which have a resolution of moderate quality and are open to access. With the development of satellites, the ICEYE mission was launched in 2018, which is a microsatellite, and the images have a resolution of up to 0.25 m. High-resolution SAR data acquired through this mission have successfully determined glacier surface velocity [23]. Additionally, a technology called Multidimensional Small Baseline Subset (MSBAS) has been developed, which can calculate the time series of north–south, east–west, and vertical ice displacement from the offset-tracking results in ascending and descending tracks. This technology can be obtained from the radar amplitude or phase information using image correlation algorithms. The glacier velocity, combined with the glacier elevation change and specific surface mass balance, can explain the response to climate change of the glacier [24–27]. The scientific community recognizes this technology as a useful tool to measure large-scale displacements caused by earthquakes, glacier motion, landslides, and other phenomena [28]. However, this technology has only been applied successfully to large-scale glaciers in polar regions. This study aims to improve the resolution of glacier velocity extraction and expand the application of MSBAS technology by utilizing it on mountain glaciers in High-Mountain Asia.

In this study, our focus is on the dynamic changes of the Siachen Glacier. The glacier surface velocity was inverted using Sentinel-1 data, which included 133 passes in the ascending and 120 passes in the descending orbit from March 2017 to December 2021. To process the data, we used GAMMA software (version 2019) to compute the offset-tracking results in range and azimuth direction, respectively [29]. Subsequently, we utilized the MSBAS methodology to calculate the displacements in three directions on the glacier surface. This technique does not depend on pixel spacing, thus resulting in a significant improvement in the temporal resolution of the displacements [30,31]. To derive the glacier surface elevation change, we registered and differenced AST14DEM images from 2008 to 2021. By combining glacier velocity, surface elevation, and morphology changes, we accurately reproduced the dynamic changes of the Siachen Glacier at a high spatiotemporal resolution. These results help to understand the causes and mechanisms of glacier movement better. The use of SAR image processing technology has enabled us to track the velocity of mountain glaciers efficiently, which is useful for monitoring glacier health and understanding the dynamic processes of glaciers.

## 2. Study Region

The Siachen Glacier is situated in the Karakoram Mountains, spanning between latitudes 35°10′ and 35°42′N and between longitudes 76°46′ and 77°25′E (Figure 1). The glacier covers an area of approximately 1063 km² with a length of almost 74 km and a thickness of around 270 m [32]. Compared to the others in the Karakoram Mountains [33], the Siachen Glacier is larger and has an altitude range from 3600 m to 7600 m with an average elevation of about 5500 m. The tongue of the Siachen Glacier is covered by debris, which constitutes approximately 3% of the entire glacier area [34]. The Siachen Glacier is the source of the Nubra River, which is one of the principal tributaries of the Indus River. The Indus converges with the Shyok River close to Sumur. Western disturbances account for the majority of precipitation in the Karakoram Mountains, and more than 50% of the snowfall takes place between November of one year and April of the following year, while the rest occurs from May to August [35].

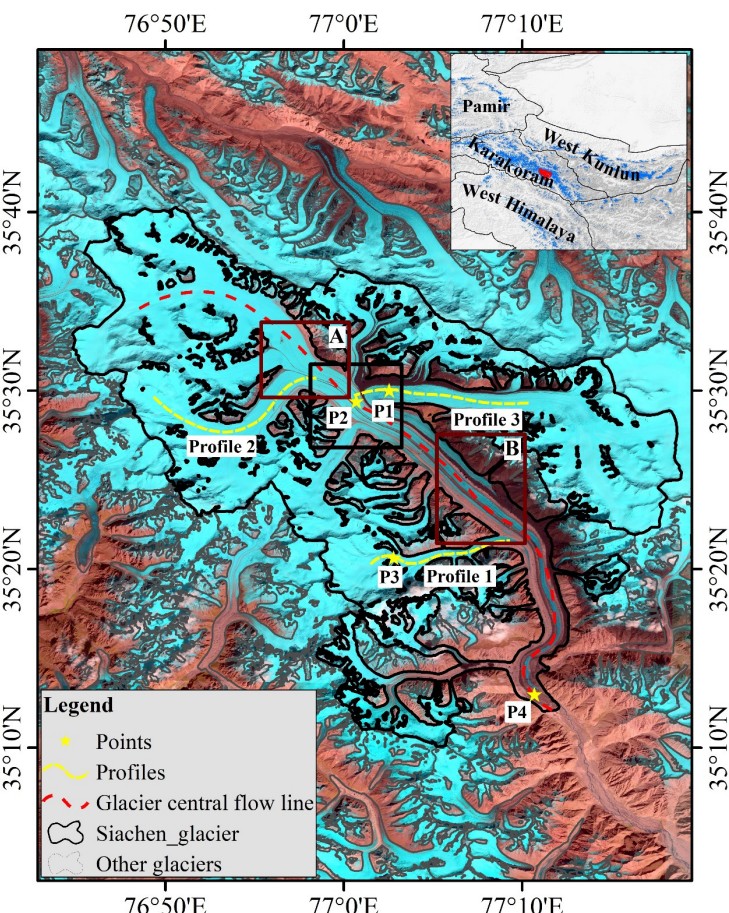

**Figure 1.** Location of the Siachen Glacier. The red line represents the glacier central flow line, and the yellow lines and pentagram symbols are profiles along three tributaries and four points (P1–P4) manually selected for further analysis of the flow velocity components. The black rectangle marks the region for further analysis of the changes of surface parallel flow (SPF) and non-surface parallel flow (nSPF) vertical flow velocity components, and the dark brown rectangles mark the region A and B for analysis of differential interferogram.

## 3. Materials

### 3.1. Sentinel-1 Data

A total of 253 Sentinel-1 images in ascending (133 images) and descending (120 images) orbits were used in this research. These images are in interferometric wide (IW) and single-look complex (SLC) mode with a spatial resolution of 2.3 and 14.9 m in range and azimuth, respectively. The data were acquired from 20 March 2017 to 24 December 2021, which

can be download from NASA Distributed Active Archive Center (DAAC) operated by the Alaska Satellite Facility (ASF). The detail information is shown in Table 1.

**Table 1.** Sentinel-1 data used in this study; θ and φ are the incidence and azimuth angle, respectively.

| Track and Period | | $\theta°$ | $\varphi°$ | Image Pairs |
|---|---|---|---|---|
| Ascending | 20170320~20211224 | 34 | 193 | 66 |
| Descending | 20170321~20211219 | 44 | 347 | 60 |

Each of the two ascending and two descending SAR images were linked together, resulting in 66 ascending and 60 descending image pairs, respectively. The number of offset results in azimuth and range is 249 and contributes to 124 SLC image pairs after the boundary correction [11,22]. GAMMA software was used to process the SAR images [29], and we produced offset results in range and azimuth for each image pair. The offset results were filtered by the Gaussian filter and then geocoded using a 30 m AW3D30 DSM with a 5 m absolute elevation accuracy; finally, we resampled them in resolution of 120 m on the common grid. The AW3D30 30 m DSM was also filtered with the Gaussian weights, and we selected a three-sigma diameter of 1 km. The topographic gradient was produced from the filtered DSM by calculating first central derivatives along the north and east directions in the Generic Mapping Tools [22]. We set the offset results in azimuth and range direction of the two different tracks as well as the topographic gradient as the input data to the MSBAS algorithm [22] to produce glacier velocity in north–south, east–west, and vertical direction, which finally are used to compose the value of glacier surface velocity.

### 3.2. Digital Elevation Model

The AST14DEM has been continuously updated since 2000, which is generated from the ASTER Level 1A image acquired by the Visible and Near Infrared (VNIR) sensor. We selected these data to calculate the glacier surface elevation change between 21 October 2008 and 30 September 2021. The DEMs in different periods were co-registries firstly using the python script demcoreg [36] to calculate the elevation difference during the period, and then, we estimated the mass balance by geodetic methods described in Section 4.3. The resolution of this DEM is 30 m, and the data are more accurate than 25 m RMSE in XYZ dimensions according to the validation testing (https://lpdaac.usgs.gov/products/ast14demv003/, accessed on 13 April 2022).

HMA DEM has three types of dataset: along-track, cross-track, and mosaics DEM, all of which have an 8-meter resolution [37]. These data are generated from images taken by ultra-high resolution commercial satellites operated by DigitalGlobe containing data from GeoEye, QuickBird, and WorldView, which span the period from January 2002 to November 2016. In addition to the mosaic data, DEMs in along-track and cross-track have individual timestamps for change detection analysis. Every along-track DEM strip was produced by in-track or along-track stereo pairs with the same orbit, and each cross-track DEM was generated from independent monoscopic image pairs, which have a suitable geometry for stereo reconstruction with a maximum time interval of 7 days. Sensors decide the absolute geo-location accuracy of the DEMs. Generally, the along-track image pairs of the WorldView and GeoEye have an accuracy of <5 m CE90/LE90 in horizontal or vertical direction, while the Quickbird-2 image pairs have an accuracy of 23 m CE90 in each direction. The relative error of DEM depends on the geometry of stereo pairs and surface slope, but it should be less than one or two meters in general [37]. At present, the first version of HMA, 8 m DEM, does not cover the whole of High Mountain Asia (HMA), but it has good coverage in the Karakoram Mountains with an acquisition time in 2015 and 2016. Therefore, we used the data as a basic reference DEM to co-register the different AST14DEM.

## 4. Methodology

### 4.1. Glacier Outlines Delineation

According to the influences of glacier outlines on the accuracy of glacier velocity extraction and mass balance, in order to improve the corresponding uncertainty assessment, glacier outlines should be delineated for the start and end years of the study period. We obtained the glacier boundaries according to the Landsat images acquired in the ablation season in 2008 and 2021. Debris-covered regions of glacier were identified based on various characteristics, such as surface textural property, local terrain, the path of supraglacial runoff, surface elevation, and velocity change [38].

### 4.2. Glacier Velocity Retrieval

According to the data sources, the remaining remote sensing inversion methods for glacier velocity can be roughly divided into two categories: optical and SAR images. Due to the special geographical location of mountain glacier, optical images are vulnerable to cloud cover. Therefore, we select SAR images as the data source for glacier velocity extraction. The retrieval process mainly includes four steps as Figure 2 shows.

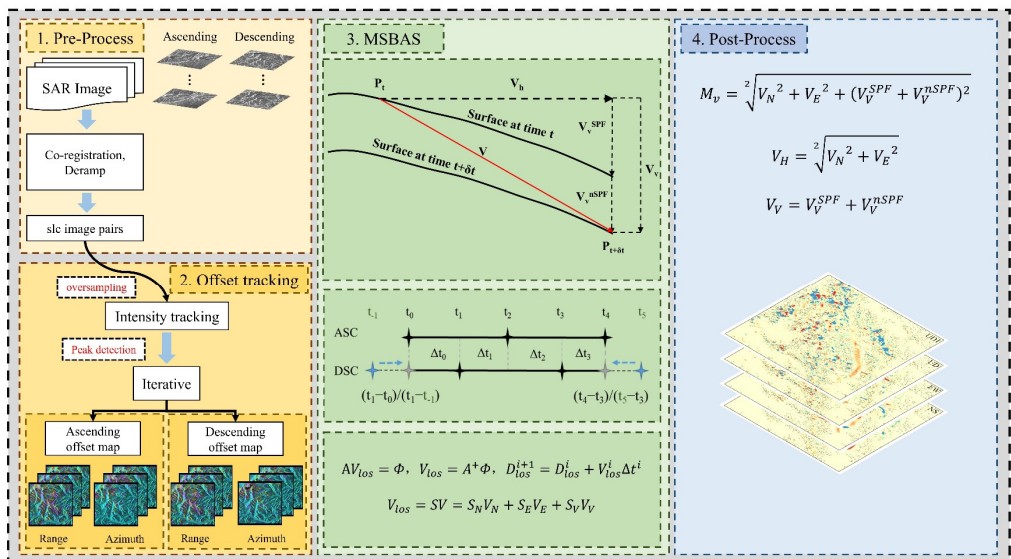

**Figure 2.** Procedure in the glacier velocity retrieval. Steps 1–4 are described in detail below. The step 1 with light yellow background is pre-process of input data, including co-registration, deramp, etc. (blue arrow sequence); then the pre-processed slc image pairs were set as the input data (black curved arrow) to the step 2: Offset tracking with dark yellow background; the step 3 with green background is MSBAS process and the step 4 with blue background is post-process for the output of step 3.

#### 4.2.1. Offset Tracking

Compared to the InSAR techniques commonly used for deformation monitoring such as landslides, surface subsidence, and earthquakes, offset tracking is not affected by the incoherence caused by rapid displacement, which is more suitable for extracting glacier velocity. There are two different techniques to compute the offset results, the intensity and the coherence tracking [9]. Both methods can use two SAR images at different times to obtain effective offset in range (along the satellite line of sight) and azimuth (along the satellite orbit) direction. The former extracts displacement information from the backscatter intensity signal, so there is no constraint on the coherence between the two images. The latter needs to calculate the offset based on the visibility of the phase and good coherence between the two images [39–41]. Glacier motion always leads to incoherence; here, we selected the intensity tracking to measure the glacier velocity. Before the offset tracking processing, we should preprocess the sentinel-1 data sequence, including

extracting the study area from initial SLC data, geocoding DEM, registration, and removing phase ramp. In detail, one image acquired in the middle of the study period was set as the reference image, and then others were co-registered to the master one for the ascending and descending tracks, respectively. The AW3D30 DEM data are used to geocode. After deramp, offset tracking processing was applied to each image pair sequentially. The spatial and temporal baselines of the image pairs are shown in Figure 3. The mean temporal resolution of SAR images in ascending and descending tracks is about 14 days in our study.

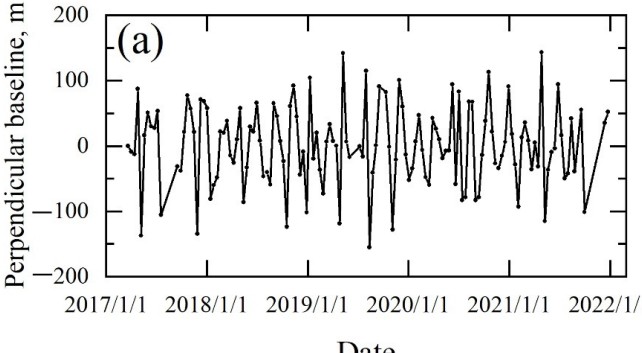
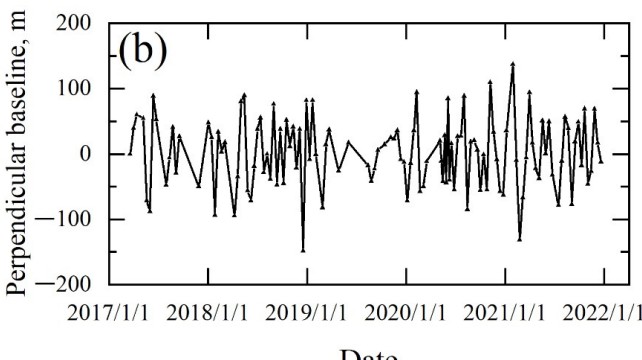

**Figure 3.** Spatial and temporal baselines between Sentinel-1 data used in this study (**a**) Ascending Sentinel−1 track 129, (**b**) Descending Sentinel−1 track 136.

The offset tracking technique obtains the offset of feature points within the search window of two SAR images through a specific matching algorithm and then calculates the corresponding ground displacement. The basic principle is to obtain the cross correlation coefficient of the feature points in two images through a cross correlation algorithm and determine the ground displacement of the feature points by searching for the maximum value of the cross correlation coefficient. The spatial domain-based intensity-normalized cross correlation algorithm utilizes the registered reference and slave images to search for two-dimensional offsets by maximizing the narrowband cross correlation function of speckle noise between two SAR images, enabling sub-pixel level matching accuracy [42]. To compute more accurate offset maps, three main steps are needed. Firstly, assuming that most areas of the two SAR images do not undergo deformation, the orbital offset can be determined by intensity cross-correlation coefficient or global optimization of phase visibility. In this study, the intensity cross-correlation coefficient is used as the indicator to estimate the orbit offset. When estimating the initial offset of an image pair, the coefficient of the range and azimuth offset polynomial functions of the entire scene image are determined using the least square method based on the local offset estimates. It is necessary to perform multi-look processing before initial offset estimation in GAMMA software. After generating the parameter file of the offset tracking processing, the offsets in range and azimuth directions can be obtained. After that, the offset of the study area can be estimated. A two oversampling process was used to reduce aliasing errors, and we firstly used a $256 \times 128$ pixel matching window and a $20 \times 5$ pixel searching step to estimate the offsets, and a $128 \times 64$ pixel sampling interval was used in the second estimation. Two steps of estimation with different sampling intervals can improve the reliability and minimize systematic biases caused by finite patch size and variable overlap. Finally, we used 0.1 as the correlation coefficient threshold to mask out unreliable offsets that may be influenced by the serious pixel distortion, less surface texture, or anomalous surface changes [11]. The real and imaginary parts of the generated results correspond to the ground displacement in the range and azimuth directions, respectively. About 6% of the glacier area does not have effective offset estimations. To improve the computational efficiency of subsequent processing, we resampled the images to 120 m by the bilinear algorithm. All offset tracking products were geocoded into WGS84 coordinates.

4.2.2. Multidimensional Small Baselines Subset (MSBAS)

The Multidimensional Small Baselines Subset (MSBAS) methodology was developed for the simultaneous processing of a large number of interferogram sequences produced by DInSAR or offset map sequences generated by offset tracking [31,43]. The offset in the range direction is sensitive to both horizontal and vertical movements, while the results in the azimuth direction are only sensitive to the offset in the horizontal direction [22,44]. The range and azimuth displacements of the ascending and descending orbit can reconstruct the components of velocity in the north–south, east–west, and vertical directions [45]. Depending on the control factors, number of data and acquisition parameters, this method can produce displacements in different dimensions. The control factors include the order of Tikhonov regularization and the value of λ and the use of terrain gradients [22]. Regularization has similar effects to low-pass filtering, and it is necessary to use regularization when the acquisition time of the ascending and descending datasets is inconsistent [43,46]. In this study, we obtained the north, east, and vertical components of glacier velocity of the Siachen Glacier with the combination of offset tracking results in range/azimuth directions from descending/ascending tracks and the terrain gradients derived from the AW3D30 DSM.

Three independent sets of differential interference results are required to calculate accurate time series of surface deformation in three directions. However, currently commonly used SAR sensors can only obtain two sets of data, namely ascending and descending. Therefore, it is necessary to introduce additional data or perform certain constraints to calculate the accurate deformation rate. As a first order approximation, glacier motion can be reduced from three degrees of freedom to two by assuming that ice flow motion occurs parallel to the glacier surface without considering the internal deformation below, allowing for all three components of velocity to be decomposed from at least one pair of ascending and descending SAR images [8]. The surface parallel flow (SPF) can be expressed as follows:

$$V_V = \frac{\partial H}{\partial X_N} V_N + \frac{\partial H}{\partial X_E} V_E \tag{1}$$

where $H$ is the terrain height, and $\frac{\partial H}{\partial X_n}$ and $\frac{\partial H}{\partial X_e}$ are the first central derivation of the north and east directions, namely terrain gradient, which can be calculated with the command in Generic Mapping Tools (GMT, version 5.4.3) based on the DEM. It is necessary to perform low-pass filtering on DEM in advance to reduce the sensitivity of certain surface features, which generally do not affect the ground motion caused by certain ground fluctuations. For example, a surface sliding in the vertical direction may be smoother than DEM data, so the degree of low-pass filtering needs to be large enough to generate the same surface slope as the total slope within the region of slide occurrence. $V_n$ and $V_e$ are the velocity components in the north and east directions.

With the above assumptions, based on the range and azimuth offset results obtained from the ascending and descending SAR images, glacier surface velocity can be decomposed into north, east, and vertical directions, which can be expressed as

$$\begin{pmatrix} \sin(\varphi_a)\sin(\theta_a)\partial t_a & -\cos(\varphi_a)\sin(\theta_a)\partial t_a & \cos(\theta_a)\partial t_a \\ \cos(\varphi_a)\partial t_a & \sin(\varphi_a)\partial t_a & 0 \\ \sin(\varphi_d)\sin(\theta_d)\partial t_d & -\cos(\varphi_d)\sin(\theta_d)\partial t_d & \cos(\theta_d)\partial t_d \\ \cos(\varphi_d)\partial t_d & \sin(\varphi_d)\partial t_d & 0 \end{pmatrix} \begin{pmatrix} V_N \\ V_E \\ V_V \end{pmatrix} = \begin{pmatrix} RO_a \\ AO_a \\ RO_d \\ AO_d \end{pmatrix} \tag{2}$$

where $\theta$ and $\varphi$ are the azimuth and incidence angles of the sensor; $RO$ and $AO$ are the offset-tracking results based on the time interval $\partial t$; and subscripts $a$ and $b$ represent ascending and descending, respectively. $V_N$, $V_E$, and $V_V$ are the velocity components in three directions.

In general, the acquisition time and interval of SAR images of ascending and descending orbit are different, and sometimes there are data gaps. In this case, calculating the velocity vectors in three directions requires assuming that no displacement deformation

occurs during the acquisition time interval of the data in the two orbits. By calculating the rate of deformation rather than the value of deformation, it is possible to allow a slight difference in the acquisition time of SAR images in different orbit. In Formula (2), three unknown quantities with four equations belong to a system of over-determined equations without exact solutions, while an approximate solution can be obtained using the least square method.

However, in most cases, an additional component is generated in the vertical direction due to the glacier surface elevation change during acquisition gap. For instance, the glacier mass lost due to glacier movement may exceed the mass gain replenished by precipitation, which leads to the elevation change of glacier surface over time. Many factors influence the boundary condition of the glacier kinematic, such as the glacier surface elevation, the specific mass balance rate, the vertical ice-particle velocity, the average density of column, the surface slope, and the velocity vector in horizontal direction [47,48]. The relationship between the change rate of glacier surface elevation and these factors are expressed as in Formula (3):

$$\dot{s} = V_V + \dot{b} - V_h \nabla s \tag{3}$$

where $\dot{s} = \frac{\partial s}{\partial t}$ is the change rate of glacier surface elevation; $V_V$ is the vertical ice-particle velocity; $V_h$ is the velocity vector in horizontal direction; $\nabla s$ is the terrain gradient; and $\dot{b} = \frac{1}{\rho}\frac{\partial b}{\partial t}$ is the specific mass balance rate, where $\rho$ is the average density of column.

The $V_V$ is the vertical component of glacier surface velocity derived from the SAR images, which is decomposed to the surface parallel flow (SPF) and non-surface parallel flow (nSPF). The velocity of SPF is equal to $V_h \nabla s$, while the velocity of nSPF is equal to $\dot{s} - \dot{b}$. So the Formula (2) can be rewritten as follows:

$$\begin{pmatrix} \sin(\varphi_a)\sin(\theta_a)\partial t_a & -\cos(\varphi_a)\sin(\theta_a)\partial t_a & \cos(\theta_a)\partial t_a & \cos(\theta_a)\partial t_a \\ \cos(\varphi_a)\partial t_a & \sin(\varphi_a)\partial t_a & 0 & 0 \\ \sin(\varphi_d)\sin(\theta_d)\partial t_d & -\cos(\varphi_d)\sin(\theta_d)\partial t_d & \cos(\theta_d) & \cos(\theta_d)\partial t_d \\ \cos\varphi_d\partial t_d & \sin\varphi_d\partial t_d & 0 & 0 \\ \frac{\partial s}{\partial X_N} & \frac{\partial s}{\partial X_E} & -1 & 0 \end{pmatrix} \begin{pmatrix} V_N \\ V_E \\ V_V^{SPF} \\ V_V^{nSPF} \end{pmatrix} = \begin{pmatrix} RO_a \\ AO_a \\ RO_d \\ AO_d \\ 0 \end{pmatrix} \tag{4}$$

where $\varphi$ and $\theta$ are the azimuth and incidence angles of the sensor. $RO$ and $AO$ are the offset-tracking results based on the time interval $\partial t$; subscripts $a$ and $b$ represent ascending and descending, respectively. $V_N$ and $V_E$ are the velocity components in the north–south and east–west directions; the $V_V^{SPF}$ and $V_V^{nSPF}$ are the SPF and nSPF components of vertical velocity; and the $\frac{\partial s}{\partial X_n}$ and $\frac{\partial s}{\partial X_e}$ are the topographic gradient in north and east directions.

Formula (4) contains four equations and four unknowns, so it is full rank, which can be further expressed as follows [22]:

$$\begin{pmatrix} A \\ H \\ \lambda L \end{pmatrix}\begin{pmatrix} V_n \\ V_e \\ V_v^{SPF} \\ V_v^{nSPF} \end{pmatrix} = \begin{pmatrix} RO \\ AO \\ 0 \\ 0 \end{pmatrix}, \begin{pmatrix} V_n \\ V_e \\ V_v^{SPF} \\ V_v^{nSPF} \end{pmatrix} = \begin{pmatrix} A \\ H \\ \lambda L \end{pmatrix}^+ \begin{pmatrix} RO \\ AO \\ 0 \\ 0 \end{pmatrix}, D_j^{i+1} = D_j^i + V_j^i \partial t^i, j = n, e, v^{SPF}, v^{nSPF} \tag{5}$$

where $A$, $H$, and $L$ are the geometry, topography, and Tikhonov regularization matrix, respectively. $\lambda$ is the parameter of regularization. $RO$ and $AO$ are the offset matrices in range and azimuth directions, and $V^i$ and $D^i$ are the glacier velocity and displacement at time $i$.

The four unknown glacier velocity components at each time were inverted using singular value decomposition according to the Formula (4), and the deformation time series of the four components were computed by numerical integration. Finally, we used linear regression to fit a straight line to the displacement time series of the four directions and derived the four linear velocity components. Because of the different time of SAR image acquisition in different tracks, the need for regularization arises. If the objective is to fill the

temporal gaps due to lacking data, we need to apply the higher order, which can smooth the results and interpolate the missing values in the temporal domain. In this study, we chose the second-degree for Tikhonov regularization and a λ of 0.01 [49].

The magnitude of the glacier velocity is calculated by Formula (3) as follows:

$$M_v = \sqrt[2]{V_n{}^2 + V_e{}^2 + \left(V_v^{SPF} + V_v^{nSPF}\right)^2} \tag{6}$$

The magnitude of the glacier velocity in the horizontal direction is calculated by the formula as follows:

$$V_H = \sqrt[2]{V_N{}^2 + V_E{}^2} \tag{7}$$

The vertical glacier velocity component can be calculated by the formula as follows:

$$V_V = V_V^{SPF} + V_V^{nSPF} \tag{8}$$

### 4.3. Estimation of Glacier Mass Balance

The geodetic method is widely used to calculate mass changes in the entire individual glacier or part of the large glacier. The total mass balance consists of surface, internal, and basal mass changes for mountain glacier generally [50], which can be expressed as follows:

$$MB_{total} = MB_{surface} + MB_{internal} + MB_{basal} \tag{9}$$

Density conversion is required in the research process, and the study period varies from several years to decades in general. Mass balance derived from the geodetic method is regarded as the volume changes ($\Delta V$) during a study period ($t_0{\sim}t_1$), calculating from two DEMs over the glacier area at $t_0$ and $t_1$ with a volume-to-mass conversion factor:

$$MB_{geodetic} = \frac{\Delta V}{\overline{S}} \times \frac{1}{t_1 - t_0} \times \frac{\overline{\rho}}{\rho_{water}} \tag{10}$$

where $\overline{S}$ is the averaged glacier area at $t_0$ and $t_1$, and $\overline{\rho}$ is the average conversion density, which is frequently used as $850 \pm 60$ kg m$^{-3}$ [51].

Many factors could cause the deviation of DEM from different sources, such as the instability of sensors, the limitation of DEM processing methods and steps, and the lack of sufficient field surveys [52]. Therefore, it is necessary to correct the spatial matching deviation of multi-source DEM data before calculating the glacier mass balance based on the geodetic method. Deviations among different DEMs include the geographical locating deviation, the bias from elevation distortion and the error related to the satellite orbit module. The elevation difference between the two DEMs contains matching deviation and glacier surface elevation change. Therefore, the influence of glacier change should be excluded before co-registration. Glacier boundaries are used to mask the DEMs, and the matching error can be estimated by the spatial offset of the non-glacial regions with the assumption that the non-glacial regions remain stable during the study periods.

### 4.4. Accuracy Assessment

4.4.1. Accuracy Assessment of the Flow Velocity Components

The uncertainty of velocity components contains the errors from MSBAS inversion and offset-tracking technique. Previous studies indicated that the former is influenced by the input data but not the suboptimal acquisition geometry of Sentinel-1 data, such as non-orthogonal orbits [11,22]. The uncertainty of each offset map is reported as 1/10~1/30 of the image pixel size of Sentinel-1 data or 0.2 and 1 m in range and azimuth directions [9]. These factors are almost independent of the accuracy of glacier velocity due to the large amounts of offset results and the large magnitude of glacier velocity. The length of the time series is more important than the precision of the individual offset map. This is similar to the precision of deformation rate derived from GNSS, which depends on the length of the

time series rather than the precision of each measurement from GNSS [22]. The average standard deviations of mean linear velocities over the entire region for north–south, east–west, and SPF/nSPF vertical components are 0.27, 0.32, 0.13, and 0.27 m a$^{-1}$, respectively. The maximum values are 5.32, 5.43, 5.47, and 5.03 m a$^{-1}$, which are relatively smaller than the value of larger glaciers in the polar region [22]. These larger values might be introduced by the velocity seasonal changes or possible surge activities.

### 4.4.2. Accuracy Assessment of the Magnitude of Linear Flow Velocity and Surface Elevation Change

The most reasonable method for the estimation of absolute accuracy is comparing the results derived from the remote sensing technique with in-situ measurements [53]. Unfortunately, this is not available for our study region and period due to the special geographical location and harsh natural conditions of mountain glaciers. On the one hand, to verify the accuracy of the composite magnitude of linear flow velocity, the glacier motion displacements at a sampling resolution of 50 m derived from multi-source data (Landsat-8, Sentinel-1/2) between 2017 and 2018 [32] are used as the reference to compare with our results. The product was processed with a cross-correlation method following the displacements of surface features or speckles and at an accuracy of about 10 m a$^{-1}$ and provided a reliable reference for our glacier velocity [24,54]. Considering the acquiring time of the products, we selected the Sentinel-1 data between 2017 and 2018 to calculate the glacier velocity using the offset tracking and MSBAS approach mentioned above. It was found that the variation trends of the glacier velocity from Millan et al. [32] and our study were consistent along the glacier central flow line of the Siachen Glacier (Figure 4), and the average STD of the velocity along glacier central flow line was 2.24 m a$^{-1}$. The correlation coefficient of the two was 0.52, and the results had a consistency in the glacier tongue. The correlation coefficient was beyond 0.80 from 80% to the terminus (Table 2), and all the results passed the significance test at 0.05 level.

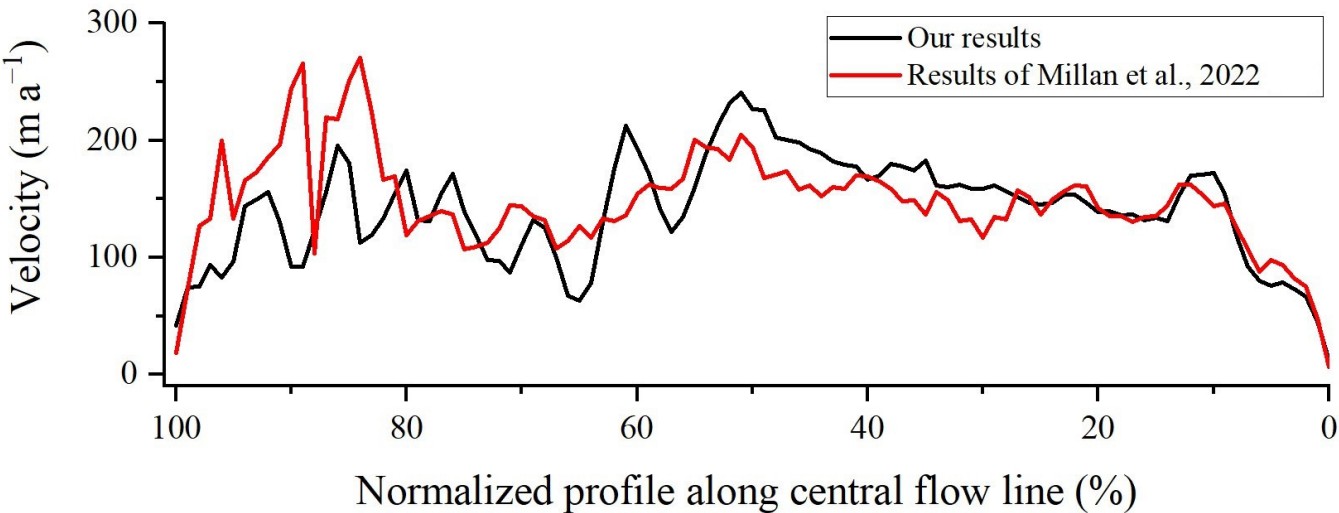

**Figure 4.** Comparison of glacier velocity along the profile of the central flow line at the Siachen Glacier between our results and Millan et al. [32].

On the other hand, the residual error over the non-glacial area with a slope smaller than five degrees is used to estimate the accuracy of the mean velocity with the assumption that non-glacial regions remain stable during the study period without deformation, which can be calculated as follows:

$$E_{off} = \sqrt{\text{MO}^2 + \text{SE}^2} \tag{11}$$

where $E_{off}$ is the uncertainty of the mean velocity of the non-glacial region, and MO and SE are the mean velocity and standard error of the MO. The SE is calculated by Formula (9):

$$SE = \frac{STDV}{\sqrt{N_{eff}}} \tag{12}$$

where $STDV$ is the standard deviation of the mean velocity in the non-glacial area, and $N_{eff}$ is the pixel number of spatial de-correlation and can be calculated by the pixel resolution and maximum distance of the spatial correlation which is generally 20 times the pixel resolution. The results show that the residual error of velocity in the non-glacial area is 5.03 m a$^{-1}$.

Image quality, registration error, and elevation residual can affect the accuracy of elevation difference. The error distribution before and after DEM registration shows that the elevation difference (MED) and NMAD of corrected DEM data have significantly decreased in non-glacier regions. ICESat-2 has been widely used for DEM accuracy evaluation due to its high precision since its launch in 2018. We selected ATL06 data on 30 September 2021 to cover the study area and evaluate the accuracy of AST14DEM data after registration. Figure 5 shows the correlation between the two different elevation data, and the results show a good agreement. The mean deviation, standard deviation, and RMSE of the elevation difference of the study area are about −15, 28, and 32 m, respectively. These can be eliminated during the registration. Hence, the uncertainty of glacier surface elevation change is evaluated using the normalized median absolute deviation (NMAD) of elevation difference in non-glacial areas. The use of median and absolute deviation to measure data dispersion makes NMAD more stable and less susceptible to the influence of outliers. Additionally, NMAD is obtained by dividing the median of the absolute deviations by the median of the whole dataset, so it is almost not affected by changes in the size of the data and the normalized result is easier to compare. The displacements in the non-glacial area with each five-degree slope bin are used to calculate the NMAD and, finally, the area weighted by the glacier area, and the result is 0.4 m a$^{-1}$ (5.2 m) from October 2008 to October 2021.

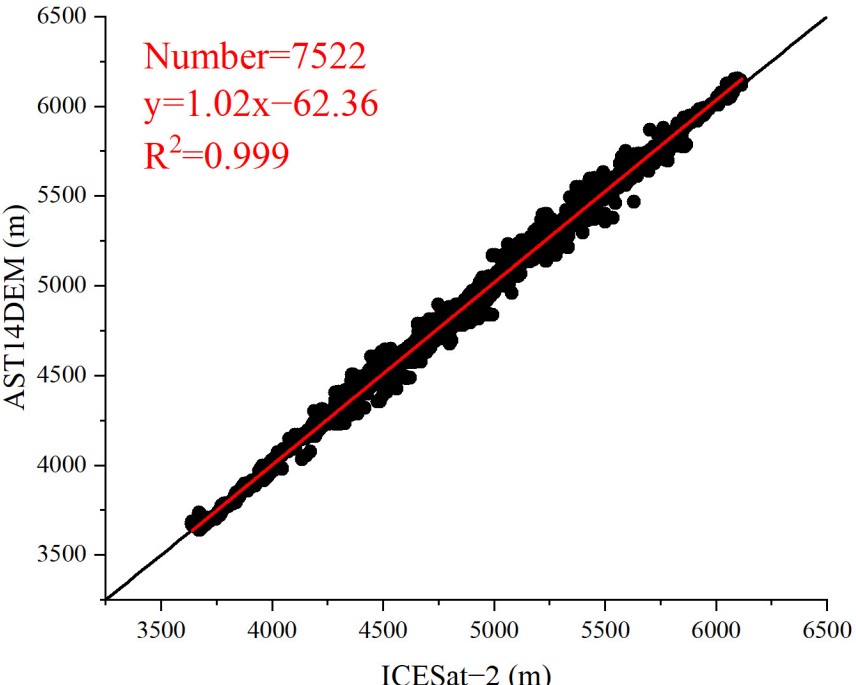

**Figure 5.** Correlation of the elevation in study area between the ICESat-2 and AST14DEM.

**Table 2.** Correlation examination of the glacier velocity along the different normalized profile of central flow line of the Siachen Glacier between our results and Millan et al. [32] All Pearson correlation coefficients passed the 0.05 significance test.

| Normalized Central Flow Line (%) | Correlation Coefficient * |
|:---:|:---:|
| 100 | 0.52 |
| 80 | 0.81 |
| 60 | 0.88 |
| 40 | 0.91 |
| 20 | 0.97 |

* Pearson correlation coefficients passed the 0.05 significance test.

## 5. Results and Discussion

The magnitude of linear flow velocity and components in three directions (north, east, and vertical) from March 2017 to December 2021 are plotted in Figure 6. Twelve glaciers have an area of >10 km$^2$. The Sherpikang Glacier, with an area of 67.87 km$^2$, has the largest average velocity (49.70 m a$^{-1}$) from 2017 to 2021. It flows with the largest velocity near the equilibrium line influenced mainly by the mass gaining and the steep slope. It also has a larger velocity at the confluence of the branches. The average surface velocity of the Siachen Glacier is 38.25 m a$^{-1}$, and the average north, east, and vertical displacement rates are $-6.06$, 13.15, and 4.28 m a$^{-1}$, respectively. The maximum value of glacier velocity in the north–south direction is 152.35 m a$^{-1}$, in the east–west direction is 335.25 m a$^{-1}$, and in the vertical direction is 166.13 m a$^{-1}$ of the Siachen Glacier.

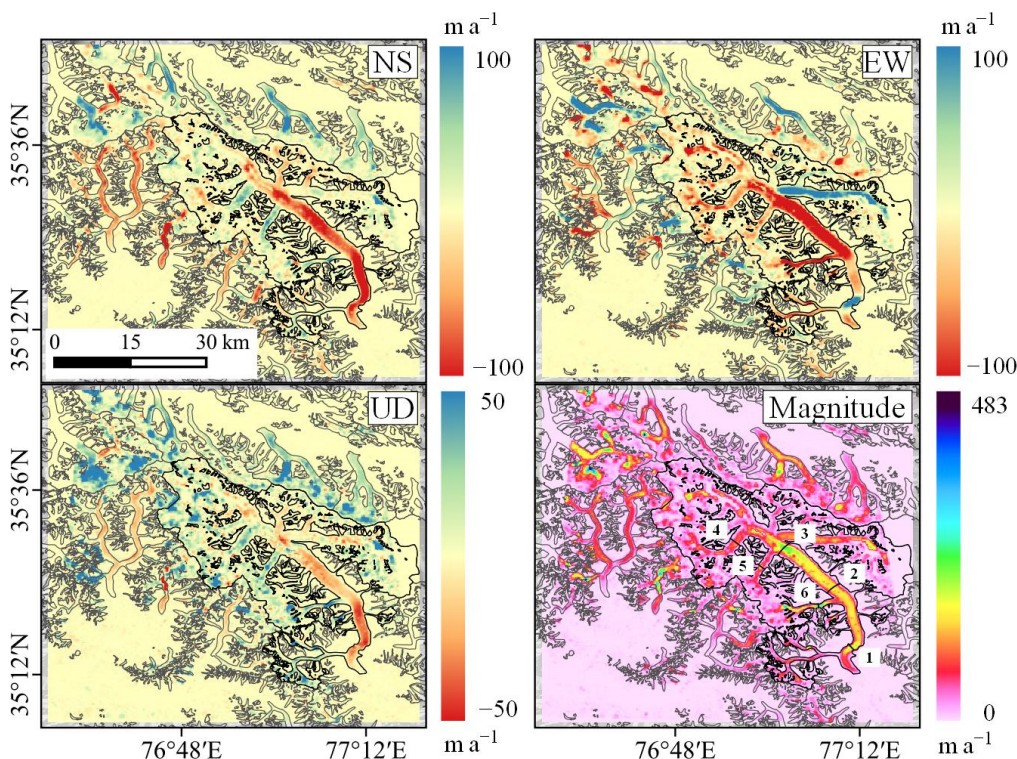

**Figure 6.** The magnitude of linear flow velocity and components in three dimensions (NS, EW, and UD) and magnitude. The black bolder lines in Magnitude are selected to analyze the distribution of glacier velocity along the glacier cross sections 1–6, cross sections 1–3 are in the glacier tongue, and 4–6 are in the tributaries.

The velocity of the Siachen Glacier follows a clear seasonal pattern, as observed in Figure 7. The maximum velocity of the glacier (0.38 m d$^{-1}$) is attained during the late spring to early summer while the minimum velocity (0.23 m d$^{-1}$) is recorded in the late ablation season. Figure 8 denotes the local differential interferogram results of the Siachen

Glacier. The results have indicated obvious interferogram fringes on the surface of the glacier from December 2019 to January 2020, which indicated that the glacier flowed slowly and experienced less decorrelation on the surface during this period. The absence of interferogram fringes from June to July 2020 suggests faster glacier movement and increased decorrelation. Notably, a slight slide in region 5 may be related to the rock avalanche in September 2010 [50]. Figure 9 depicts the distribution of glacier velocity along the cross sections shown in Figure 6, and the outcomes have shown that the velocities are most prominent at the central flow line, declining proportionally towards zero to the sides. The velocity profile (Figure 9f) along the cross Section 6 marked in the Figure 6 is on a relatively narrower tributary of the glacier, where the small width causes a large velocity near both sides of the valley; this may be related to the glacial erosion and steep terrain.

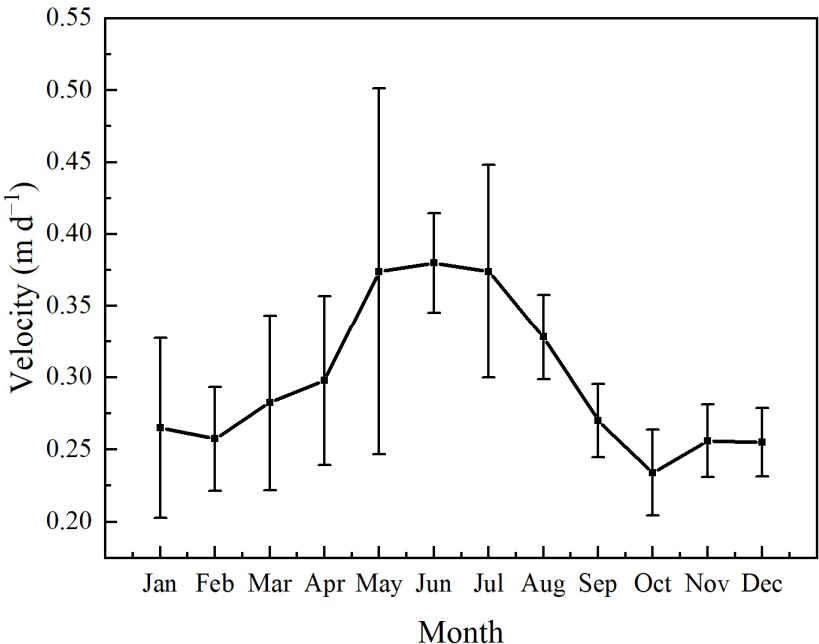

**Figure 7.** The monthly averaged velocity of the Siachen Glacier during the period of 2017–2021.

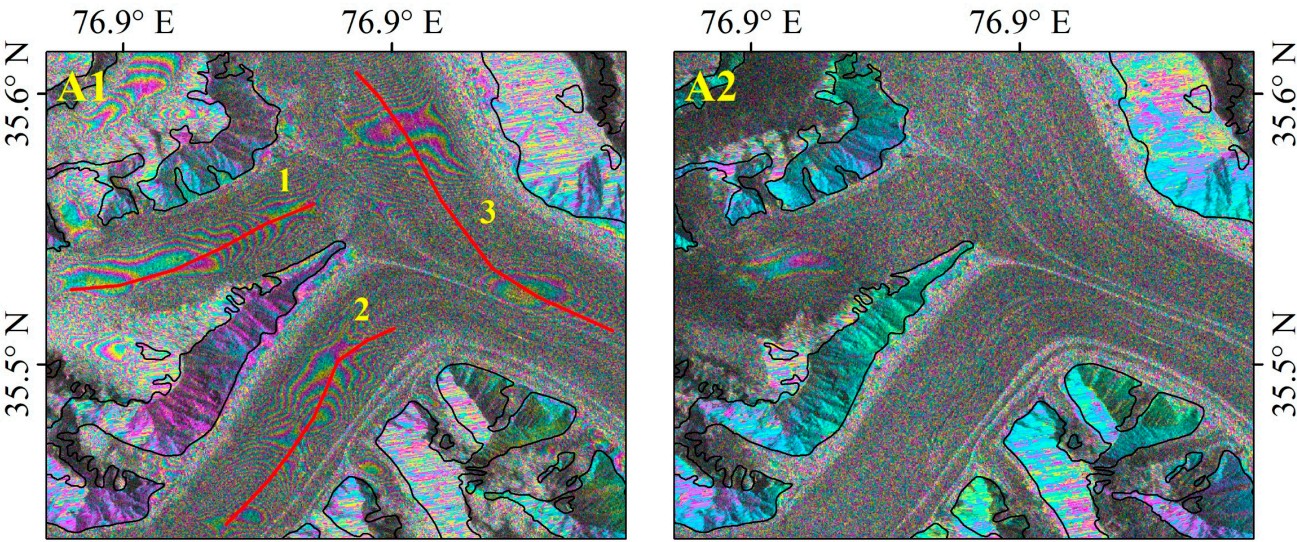

**Figure 8.** *Cont.*

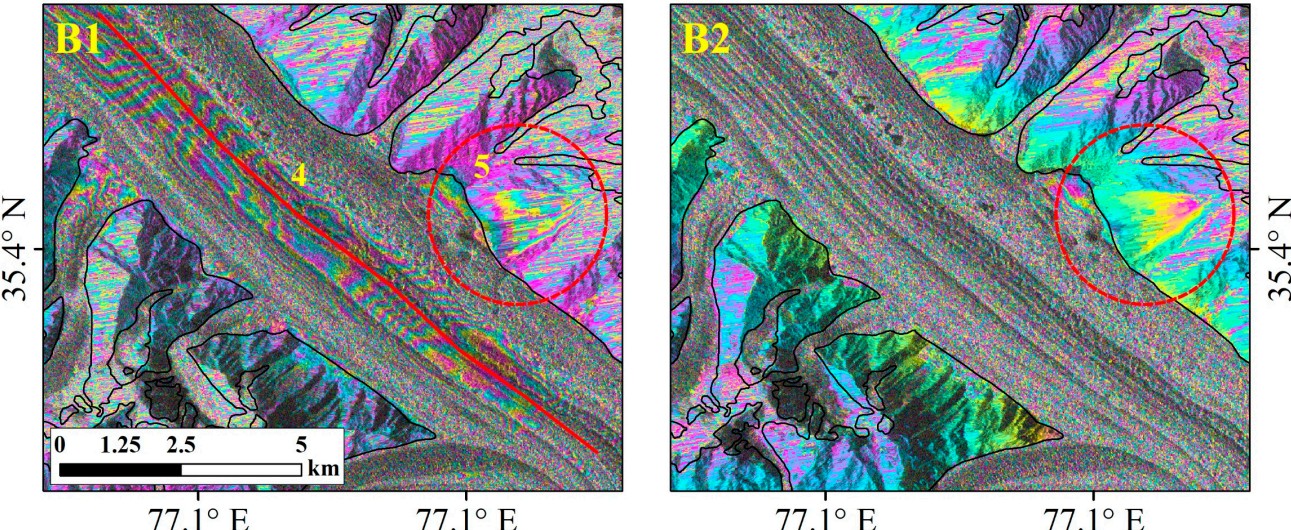

**Figure 8.** The differential interferogram of the Siachen Glacier in period 1: 20191223–20200104 and 2: 20200912–20200924: (**A1,A2**) are for region A in period 1 and 2, (**B1,B2**) are for region B in period 1 and 2, region A and B are marked in Figure 1. The red lines are the profiles for detail analysis, and the red circle indicates the area of the rock avalanche. The base map is the intensity map.

The components of displacement in different directions of points 1~4 are shown in Figure 10. The results indicate that the horizontal velocity of the glacier is generally larger than the vertical component and controls the magnitude of the glacier surface velocity. The displacements in the east–west direction of point 1 is much larger than the displacement time series in other directions, which is influenced by the direct west flow direction and gentle slope of the tributary. Vertical nSPF components of point 2 and 4 experience a seasonal pattern. However, there is no obvious seasonal change in the four components of point 3, which is possibly related to the narrower valley.

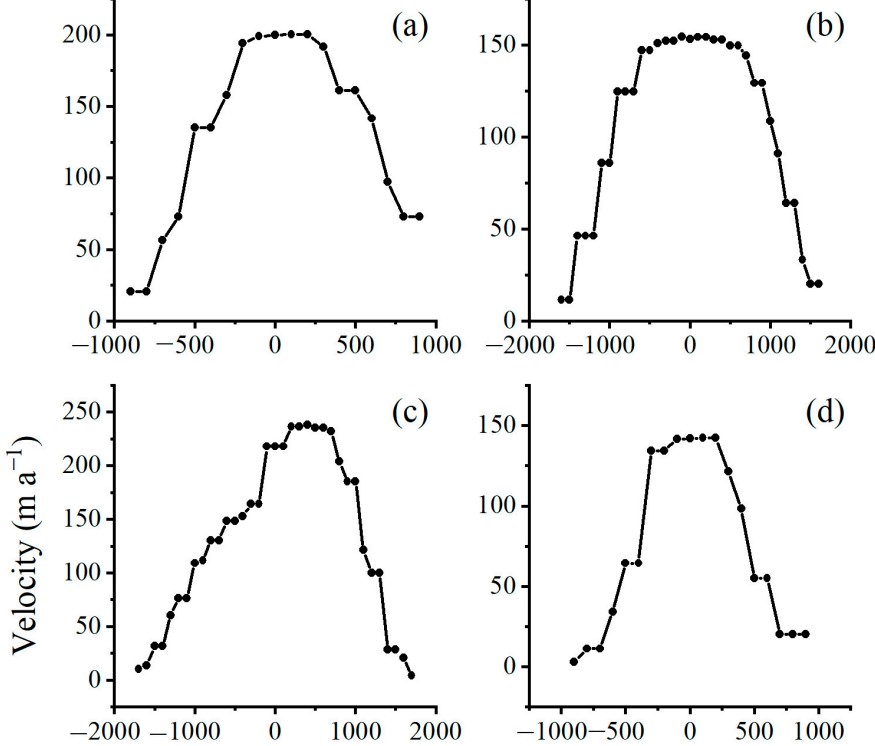

**Figure 9.** *Cont.*

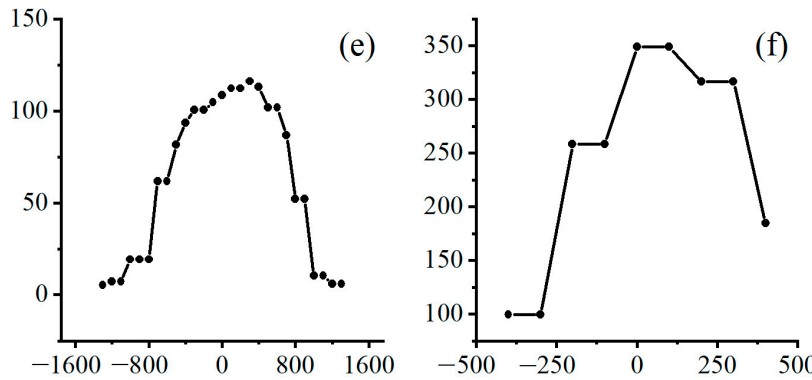

**Figure 9.** The distribution of average glacier velocity along the glacier cross section of the Siachen Glacier from 2017 to 2021. (**a–f**) represent the cross section 1–6 shown in Figure 5.

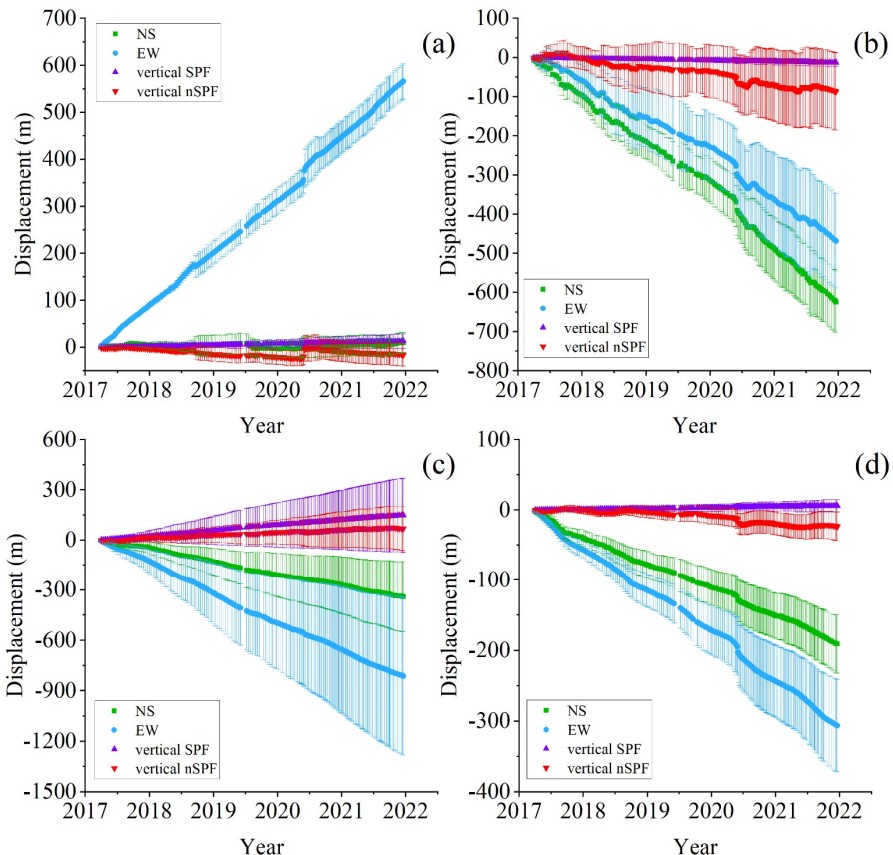

**Figure 10.** Flow displacements time series for points 1–4 in 2 × 2 pixel, whose locations are shown in Figure 1: (**a–d**) correspond to points 1~4, respectively.

The SPF and nSPF components are shown in Figure 11. The magnitude of the SPF component changes along the mainly downward flow direction moderately. By comparison, the magnitude and direction of the nSPF component experience major changes in some regions. In general, the velocity vector at the glacier surface is downward sloping in the accumulation region and upward sloping in the ablation region, which can be found in the SPF component of the Siachen Glacier. However, there is a downward SPF component at the terminus, which may be related to the mass gain in this region. The nSPF component is related to the variations of strain rates and mass balance and can be separated from the vertical component according to the kinematic boundary condition at the glacier

surface [48,49]. Glaciers with a constant accumulation rate and no melting have a fixed density profile related to the depth under the glacier surface. According to Sorge's Law [55], the velocity component in the vertical direction ($V_V$)—that is, the vertical ice-particle (pole) velocity—can be calculated as Formula (11) with an assumption that the thickness change can be converted to the equivalent mass balance by multiplying by the averaged density of glacier ice column:

$$V_V = V_h \nabla s + \dot{s} - \dot{b} \tag{13}$$

where the $V_V = V_V^{SPF} + V_V^{nSPF}$, the $V_V^{SPF} = V_h \nabla s$, and the $V_V^{nSPF} = \dot{s} - \dot{b}$.

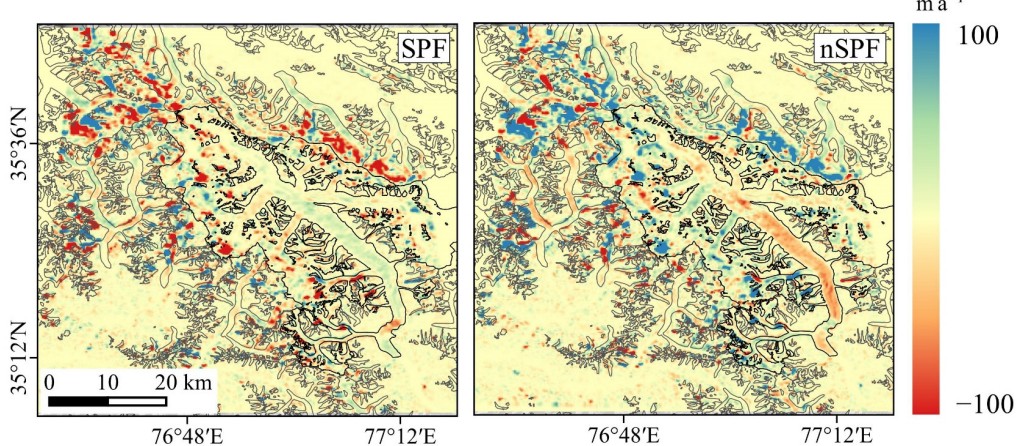

**Figure 11.** The mean SPF and nSPF velocity components separated from vertical displacement from 2017 to 2021.

We analyzed the nSPF vertical velocity component and elevation change along the glacier surface central flow line, and the results showed a significant correlation at the 0.05 level with a correlation coefficient of 0.56.

The distribution of the two components in the vertical direction is further analyzed in a tributary junction region of the Siachen Glacier with large velocity variation. The largest magnitude in velocity is located at the middle part of the selected region while values in the other parts are relatively smaller (Figure 12b). The largest nSPF component is observed at the junction of the right tributaries, where the SPF component is relatively small. This is due to the shallow topography, i.e., the $\nabla s$ is near zero, and the vertical flow velocity is mainly influenced by changes of strain rates but not sliding introduced by gravity. The nSPF vertical flow velocity is larger in the ablation region than the value in the accumulation region of the Siachen Glacier (Figure 11), which is related to the intense melting in the tongue. This indicates that the pattern of vertical motion is not uniform in the valley.

The glacier surface elevation changes during 2008~2021 is shown in Figure 13. The surface elevation change of the entire Siachen Glacier is $0.12 \pm 0.40$ m a$^{-1}$, and the corresponding mass balance is $0.07 \pm 0.23$ m w.e. a$^{-1}$. The thickness of the central flow line decreased at a rate of $-1.04$ m a$^{-1}$, while there was a significant increase ($3.27$ m a$^{-1}$) in tributary thickness where profile 2 is located (Figures 13 and 14). Negi et al. [35] calculated the average mass balance of the Siachen Glacier from 1986 to 2018, and the result is $-0.11 \pm 0.24$ m w.e. a$^{-1}$, which suggests that the Siachen glacier experienced a more intensive mass loss before 2008, and began to experience slight mass loss or even mass gain recently. Furthermore, the distribution of glacier surface elevation changes in elevation bins (Figure 15) indicated that the Siachen Glacier surged during the study area. Both the elevation changes within the entire range of the Siachen Glacier and the main part of the glacier tongue indicated that this glacier surged in the area at an altitude of about 4800 m of glacier tongue; then, the glacier mass was

rapidly transformed to the area below at an altitude of about 3800 m. These resulted in a regional thickening at the terminus.

　　A regional surge happened at the region marked by the red circle in Figure 13, with typical characteristics such as that the elevation of upper part decreased while the value of downward increased obviously. The glacier surface elevation change (Figure 14) and the changes of flow velocity (Figure 16) of the profiles show good agreement that mass gain indicates velocity increase while mass loss causes velocity decrease. Hence, the flow velocity derived from Sentinel-1 data in ascending and descending tracks using offset tracking and MSBAS technique is an effective factor in monitoring the glacier health, while the nSPF vertical flow velocity component can compensate for the lower temporal resolution of the mass balance change derived from the geodetic method. Mass balance can influence the stability of glaciers, so the tributaries of profiles 2 and 3 may surge in the future.

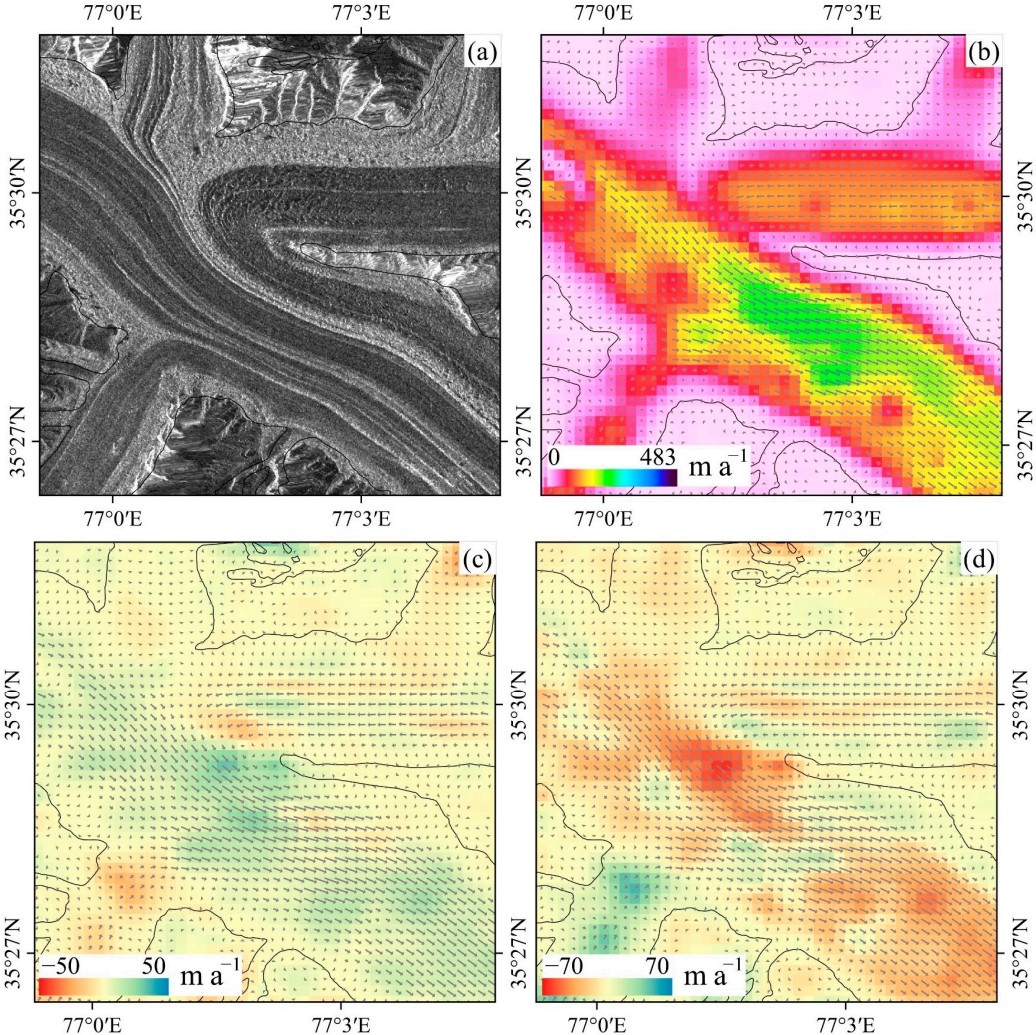

**Figure 12.** Intensity image (**a**), velocity magnitude (**b**), SPF (**c**), and nSPF (**d**) components for the region marked in Figure 1, where several tributaries join into the trunk. Vectors indicate the direction and magnitude of velocity in the horizontal direction.

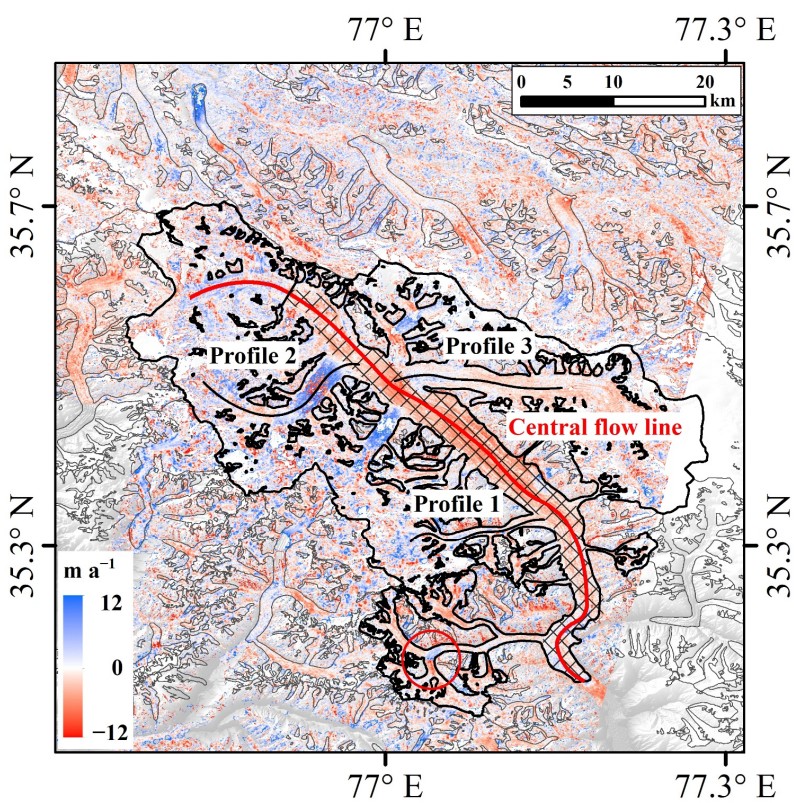

**Figure 13.** The surface elevation changes of glaciers between 2008 and 2021 derived from AST14DEM. The central flow line of the Siachen Glacier is marked in the red line; the profiles of tributary are marked in black lines; the main part of the glacier tongue is marked in crosshatch; and the red circle marked a glacier surge during the period.

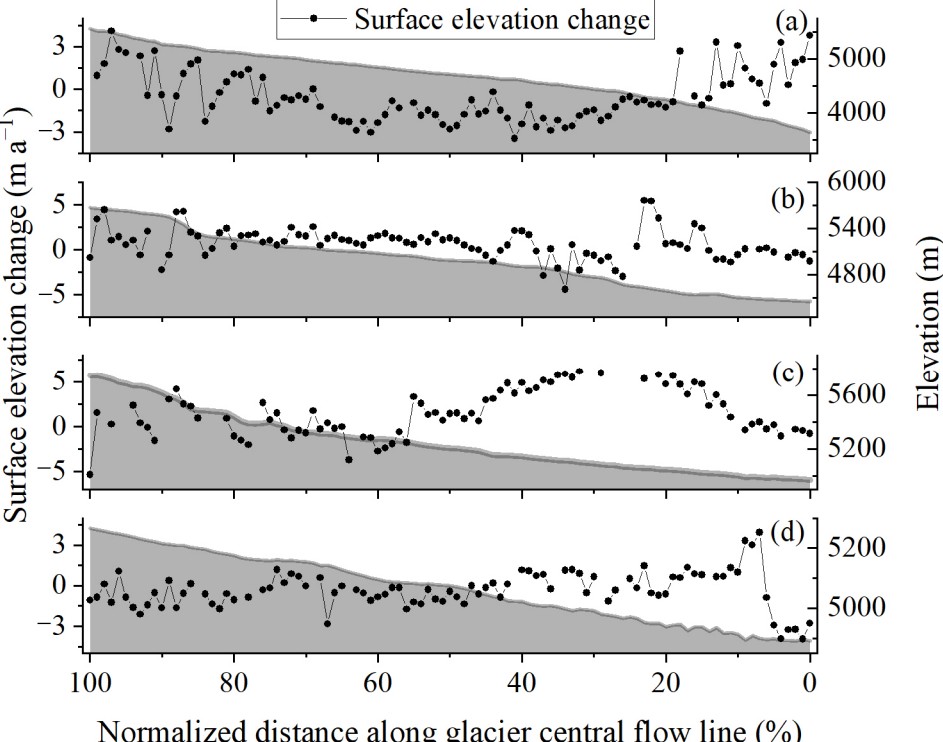

**Figure 14.** Normalized glacier surface elevation changes along the central flow line (**a**) and profiles 1~3 (**b–d**), which are marked in Figure 12.

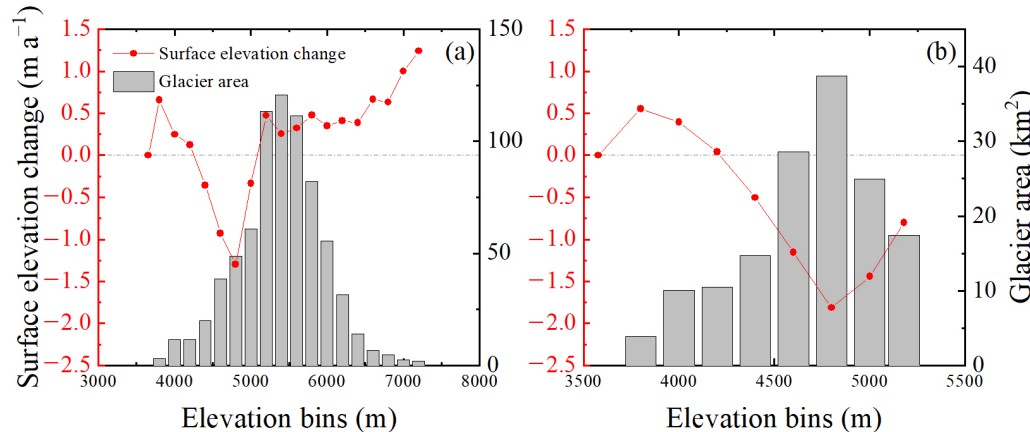

**Figure 15.** Glacier surface elevation change distribution in elevation bins of the whole Siachen Glacier (**a**) and its main part of the glacier tongue (**b**).

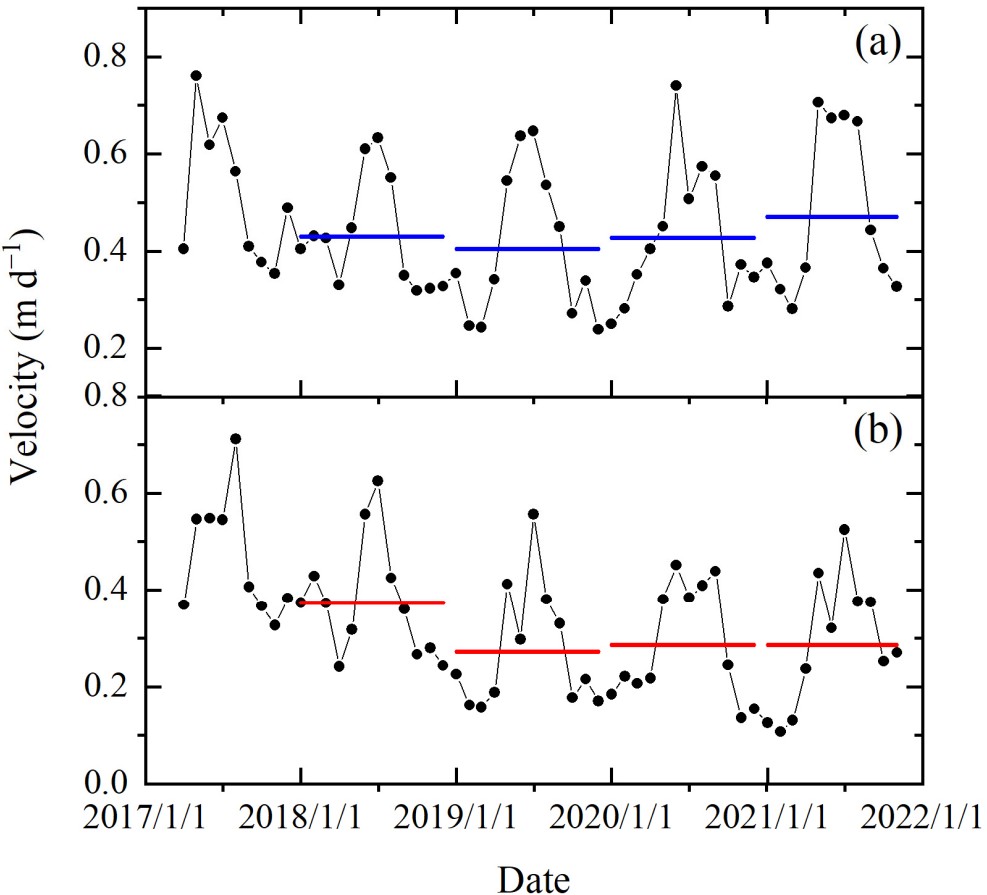

**Figure 16.** Changes of averaged velocity along profiles 2 (**a**) and 3 (**b**), which are marked in Figure 12.

## 6. Conclusions

The Siachen Glacier, which covers an area of 1063.46 km$^2$, is situated in the Karakoram Mountains. The mass balance of this glacier between 2008 and 2021 is 0.07 ± 0.23 m w.e. a$^{-1}$, and a significant regional difference is observed throughout the entire glacier region. Mass loss exists in the lower part of the glacier as opposed to mass gain in certain upper tributaries. Currently, mass balance derived from DEM difference has relatively lower temporal resolution, always in the decade scale. Velocity is a significant metric to assess the health of the glacier at a more granular level. Time series of flow velocity components are obtained with the offset tracking and MSBAS technique, which rely on Sentinel-1 IW SLC

images in both ascending and descending tracks, together with DEM data, to reconstruct the motion of the glacier. The average velocity of the Siachen Glacier from 2017 to 2021 was 38.25 m a$^{-1}$, and no substantial variation was observed during the study period. The monthly fluctuations in glacier velocity reveal a seasonal pattern that the maximum velocity occurs between the late spring to early summer, and the minimum one is found during the late ablation season. Based on the interplay between mass balance and velocity changes, it is possible that the two branch glaciers on the southwestern side of the Siachen Glacier will surge in the future. The strain rate and mass balance process are correlated with the nSPF component, which is derived from the vertical flow velocity and has an average resolution of 14 days, and can be employed to study glacier dynamics. These results can compensate for the inferior temporal resolution of geodetic mass balance and contribute vital parameters to the glacier dynamics model.

**Author Contributions:** Conceptualization, N.W. and Q.L.; methodology, N.W. and Q.L.; data processing, Q.L.; writing—original draft preparation, Q.L.; writing—review and editing, N.W.; supervision, N.W.; project administration, N.W.; funding acquisition, N.W. All authors have read and agreed to the published version of the manuscript.

**Funding:** This research was supported by the National Natural Science Foundation of China (42130516) and the Second Tibetan Plateau Scientific Expedition and Research Program (2019QZKK020102).

**Data Availability Statement:** Not applicable.

**Acknowledgments:** We thank the European Space Agency (ESA) and Alaska Satellite Facility (ASF) for providing Sentinel-1 and 2 data, NASA for providing long-time series AST14DEM, and Shean for HMA 8 m DEMs and open source demcoreg python scripts (https://github.com/dshean/demcoreg, accessed on 1 May 2022). We thank the powerful InSAR processing software: GAMMA and MSBAS open-source software from Samsonov et al. (https://insar.ca/multidimensional-small-baseline-subset/, accessed on 20 July 2022). Finally, we would like to give special thanks to Sergey Samsonov for his patience in answering the questions we encountered during the data processing with MSBAS software (version 2019).

**Conflicts of Interest:** The authors declare no conflict of interest.

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
