# Peer review of "Mountain Glacier Flow Velocity Retrieval from Ascending and Descending Sentinel-1 Data Using the Offset Tracking and MSBAS Technique: A Case Study of the Siachen Glacier in Karakoram from 2017 to 2021"

_remotesensing, doi:10.3390/rs15102594_

Round 1

Reviewer 1 Report

This paper is focused on the study of mountain glacier flow velocity retrieval. But the innovation of the proposed methodology is too limited and does not show significant advancement in this kind of studies. It pays more attentions on the results analysis not the method. So in my opinion, it is not deserved to be published in RS.
 Additionally, the paper should be revised for more clearly in the following aspects,
1) "Depending on the control factors, number of data and acquisition parameters, this method can produce displacements in different directions" . As far as this study is concerned, what are those control factors and parameters?
2) What’s the relationship between MSBAS and offset tracking? And the method used in this study should be introduced in details, especially the MSBAS technique.
3) What are the results from MSBAS and offset tracking respectively? 
4) All the variates in formula (1) should be explained respevtively.
5) All the figures in this paper are not clear, where the contents are fuzzy. 
6) The acronyms should be expanded when they are mentioned at the first time, such as SPF?

Reviewer 2 Report

very interesting article. I would just emphasize to assure readers that the methods error halo is smaller than the result changes to ensure adequate analyses of the results. I would recommend empahzing this point.

Author Response

Thank you very much for your affirmation of this study. We have provided a more detailed and complete explanation of the methods used in this study in order to improve the readability of the article. For the error analysis of this study, the average standard deviation of velocity components in three directions is used to evaluate the precise of the inversion results of MSBAS. In terms of the magnitude of glacier surface velocity composed from the components, the average deformation rate and its standard deviation in non-glacier regions are statistically analyzed, and the precise is ultimately obtained based on the error transfer theory. The statistical results of these two evaluating methods are significantly smaller than the velocity of the glacier region. In addition, we compared our results with others of the same period, and they showed good consistency on the glacier central flow line, especially in the middle and lower parts of the glacier tongue. Therefore, it is feasible to use MSBAS technique to extract high spatiotemporal resolution surface velocity of mountain glacier in the High Mountain Asia.

Reviewer 3 Report

The paper titled "Mountain glacier flow speed retrieval from ascending and maleding Sentinel-1 data using the MSBAS technology: a case study of the Siachen Glacier in Eastern Karakoram from 2017 to 2021" is a well written and interesting case study on mountain glacier monitoring. The method described in detail allows for its faithful reconstruction. An interesting solution is the use of MSBAS software for data processing.
However, there are some small errors and shortcomings that please complete/correct:
1) please lay out all the drawings in the article, including the form of some of the drawings is illegible.
2) In the introduction, please expand on the remote sensing techniques listed in lines 52-54. Give their advantages and disadvantages and get the effect. It is also worth checking the literature for examples of the use of the so-called microsatellites, including ICEYE.

Round 2

Reviewer 1 Report

Thank you for your careful revision. Based on your revised manuscript, I think you are only using the displacements form ascending and descending orbits by using offset track method. MSBAS is a method based on SBAS, but you really do not use it. Therefore, the term "MSBAS" should be removed from your manuscript and the focus should be on the geometric relationship between the displacement of the offset orbit and the displacement in the north-south, east-west and vertical directions. Therefore, I believe it is recommended to resubmit it in a new manuscript form.

In addition, the English writing of this paper should be greatly improved.

Reviewer 3 Report

I accept article in present form.

Author Response

Dear reviewer,

Thank you for approving this study and its previous revisions. In this revision, we significantly revised the wording, including spelling, grammar, phrasing, and more to improve readability. We made significant modifications, focusing on the abstract, introduction, and conclusion sections. These revisions enhance the academic style and make the text more accessible to readers.

Kind regards,

Qian Liang